# Spontaneous voltage and persistent electric current from rectification of electronic noise in cuprate/manganite heterostructures

Mathias Soulier[1], Shamashis Sengupta[2], Yurii G. Pashkevich [1,3], Roxana Capu [4], Ryan Thompson [1], Jarji Khmaladze[1], Miguel Monteverde[5], Louis Dumoulin[2], Dominik Munzar[6], Christian Bernhard [1] ✉ & Subhrangsu Sarkar [1] ✉

Non-reciprocal transport in solids under time-reversal symmetry is of great current interest. Here we show that $YBa_2Cu_3O_7$(YBCO)/ $Nd_{0.65}(Ca_{0.7}Sr_{0.3})_{0.35}MnO_3$(NCSMO) multilayers are promising candidates. By rectifying environmental electromagnetic fluctuations, they generate a spontaneous voltage of tens of millivolts, that can drive a persistent current across external circuits. The underlying ratchet-type potential presumably originates from the complex domain state of the NCSMO layers which host several nearly degenerate magnetic, electronic and polar orders. Particularly important appears to be the competition between a charge/orbital ordered majority phase with polar moments and a nonpolar ferromagnetic minority phase. A central role is also played by the adjacent YBCO layers that are too thin (≤10 nm) to fully screen the electric fields emanating from the NCSMO layers. These multilayers are useful for energy harvesting over broad temperature and magnetic field ranges, and for tunable multifunctional memory devices that are responsive to magnetic fields, electric currents, and electromagnetic radiation.

The breaking of inversion symmetry in conducting materials, on microscopic and mesoscopic scales, can lead to non-reciprocal electric transport and a directional current flow that can be exploited for various device applications. It is governed by the principle that charge carriers experience unequal non-linear impedances for opposite directions of flow, leading to the generation of a d.c. voltage and higher harmonic frequencies in response to an a.c. current. Such properties have been reported in materials with a non-centrosymmetric crystal structure[1–3], systems hosting magnetic order[3–5], macroscopic chiral conductors[6], and in devices based on superconductors[7–13]. A remarkable consequence of the non-reciprocal response, reported in some superconductor-based systems[14,15], is that these are efficient for

harvesting electricity by rectification of environmental electromagnetic fluctuations into a d.c. voltage. However, in most cases, the spontaneous voltage (SpV) effect occurs only at finite magnetic fields in close vicinity of the superconducting critical field.

Here, we report the generation of a spontaneous d.c. voltage in YBCO/NCSMO heterostructures that is unusually large, persists over a wide range of temperature and magnetic fields, exhibits a magnetic memory effect, and can be used to drive a persistent current. Notably, this SpV is a time-reversal-even phenomenon which does not require an external magnetic field and is symmetric upon its sign change. We provide evidence that the underlying mechanism involves a ratchet-type potential that breaks inversion symmetry on a mesoscopic scale

[1]Department of Physics and Fribourg Center for Nanomaterials, Chemin du Musée 3, University of Fribourg, Fribourg, Switzerland. [2]Université Paris-Saclay, CNRS/IN2P3 IJCLab, Orsay, France. [3]O. Galkin Donetsk Institute for Physics and Engineering NAS of Ukraine, Kyiv, Ukraine. [4]West University of Timisoara, Faculty of Physics, Timisoara, Romania. [5]Université Paris-Saclay, CNRS Laboratoire de Physique des Solides, Orsay, France. [6]Department of Condensed Matter Physics, Faculty of Science, Masaryk University, Brno, Czechia. ✉e-mail: christian.bernhard@unifr.ch; subhrangsu.sarkar@unifr.ch

and originates from a complex domain state within the NCSMO layers that is composed of a mixture of nearly degenerate phases with distinct electronic, polar and magnetic properties.

## Results

### Magnetic and superconducting transition in N10Y7-SL

Multilayers from $YBa_2Cu_3O_7$ (YBCO) and $Nd_{0.65}(Ca_{0.7}Sr_{0.3})_{0.35}MnO_3$ (NCSMO) have been grown with pulsed laser deposition (PLD) on LSAT substrates as detailed in refs. 16–18. and in the methods. Figure 1a shows a sketch of a superlattice with ten repetitions of NCSMO(10 nm)/YBCO(7 nm) bilayers (N10Y7-SL) and an additional topmost NCSMO layer that has been most intensively studied. The transport experiments were made in four-probe geometry, either with extended line contacts made from silver paint, as sketched in Fig. 1a, or with point-like contacts made by wire-bonding. As sketched in Fig. 1a, an in-plane magnetic field has been applied perpendicular to the direction of

the measured voltage or current signals. Some of the measurements discussed in the following have been performed on corresponding NCSMO/YBCO bilayer or NCSMO/YBCO/NCSMO trilayer samples.

Figure 1b shows an XRD scan of the N10Y7-SL along (00 l) that reveals Bragg peaks characteristic of the YBCO and NCSMO structures and confirms their epitaxial growth. The obtained $c$-axis lattice parameter of the YBCO layers of $c = 1.164$ nm is characteristic of a fully oxygenated phase with close to optimally hole doped $CuO_2$ layers. The inset highlights a characteristic superlattice modulation that is superimposed on the Bragg peaks and testifies to the high quality and sharpness of the interfaces and a homogenous lateral layer thickness.

The R-T curves of the N10Y7-SL in Fig. 1c show a reasonably sharp superconducting transition with an onset temperature $T_c^{ons} \approx 80$ K and a midpoint $T_c^{mid} \approx 75$ K. They have been measured at a rather high current of 1 mA that suppresses the SpV effect discussed below and restores the typical metallic and superconducting response of the

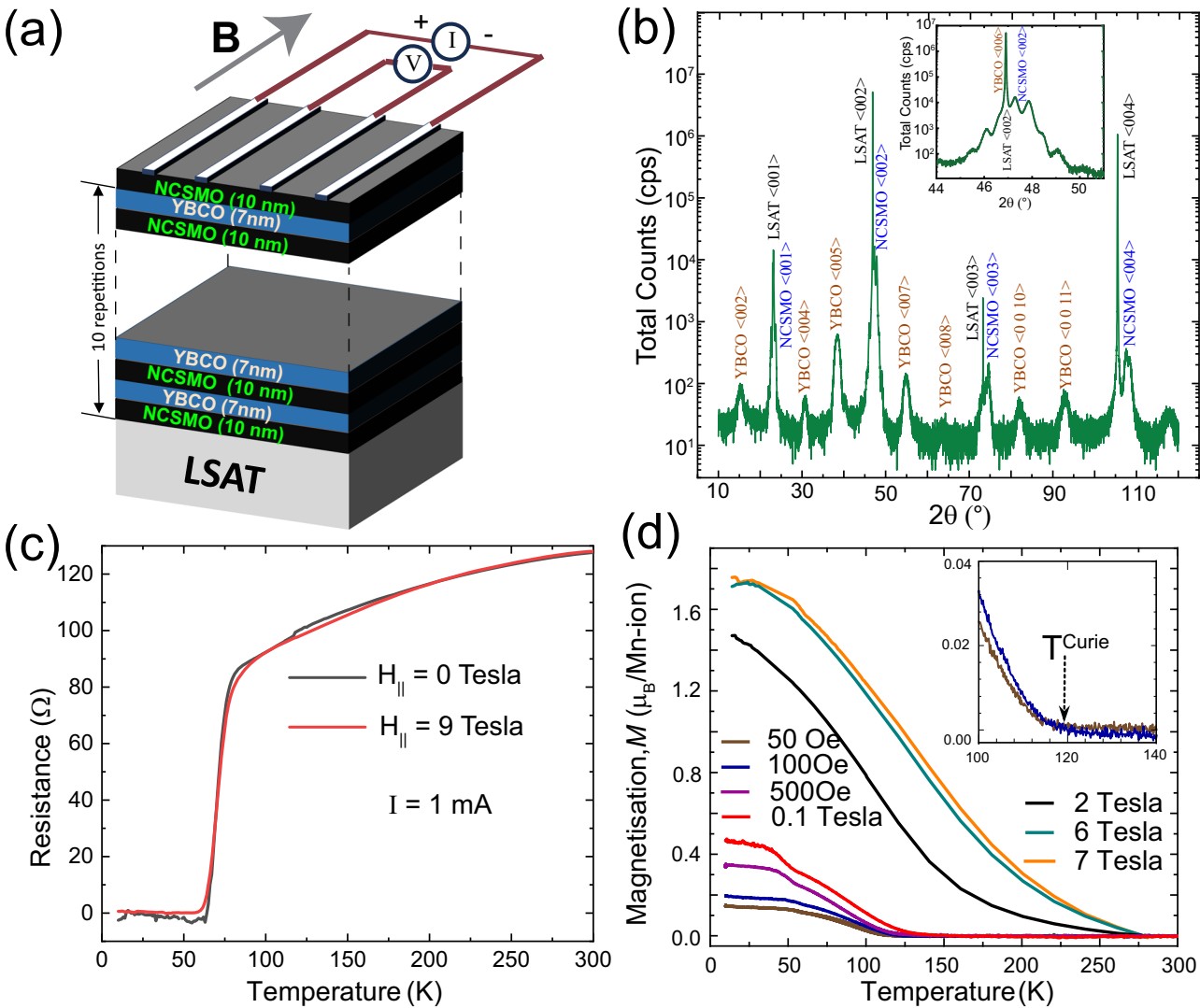

**Fig. 1 | Schematic representation and characterization of the NY-superlattice. a** Sketch of a $YBa_2Cu_3O_7$-$Nd_{0.65}(Ca_{0.7}Sr_{0.3})_{0.35}MnO_3$ (YBCO-NCSMO) superlattice with a NCSMO layer on top. Also shown is the geometry of the line contacts that are made with silver paint on the NCSMO top layer and the direction of the applied magnetic field. **b** XRD pattern along the (0 0 l) direction of the SL showing pronounced Bragg-peaks that testify to the structural quality and the epitaxial growth of the YCBO and NCSMO layers. Inset: Magnified view of a Bragg-peak with a clear superlattice modulation that is characteristic of sharp interfaces and a laterally homogeneous layer thickness (reproduced from ref. 18). **c** R-T curves in zero and 9T

for an applied current of 1 mA which suppresses the SpV signal and reveals a superconducting transition with an onset temperature of $T_c^{ons} \approx 80$ K. **d** M-T curves of the superlattice with the magnetic field applied parallel to the layers. The low field curves show a weak FM signal with an onset temperature $T^{Curie} \approx 110$–120 K, as detailed in the inset (Reprinted Figs. 1 and 2a with permission from [R. Gaina, S. Sarkar, M. Soulier, J. Khmaladze, E. Perret, E. Weschke, and C. Bernhard, Phys. Rev. B, 104, 174513 (2021)] Copyright (2021) by the American Physical Society.). Some additional data has also been added to 1d of this manuscript.

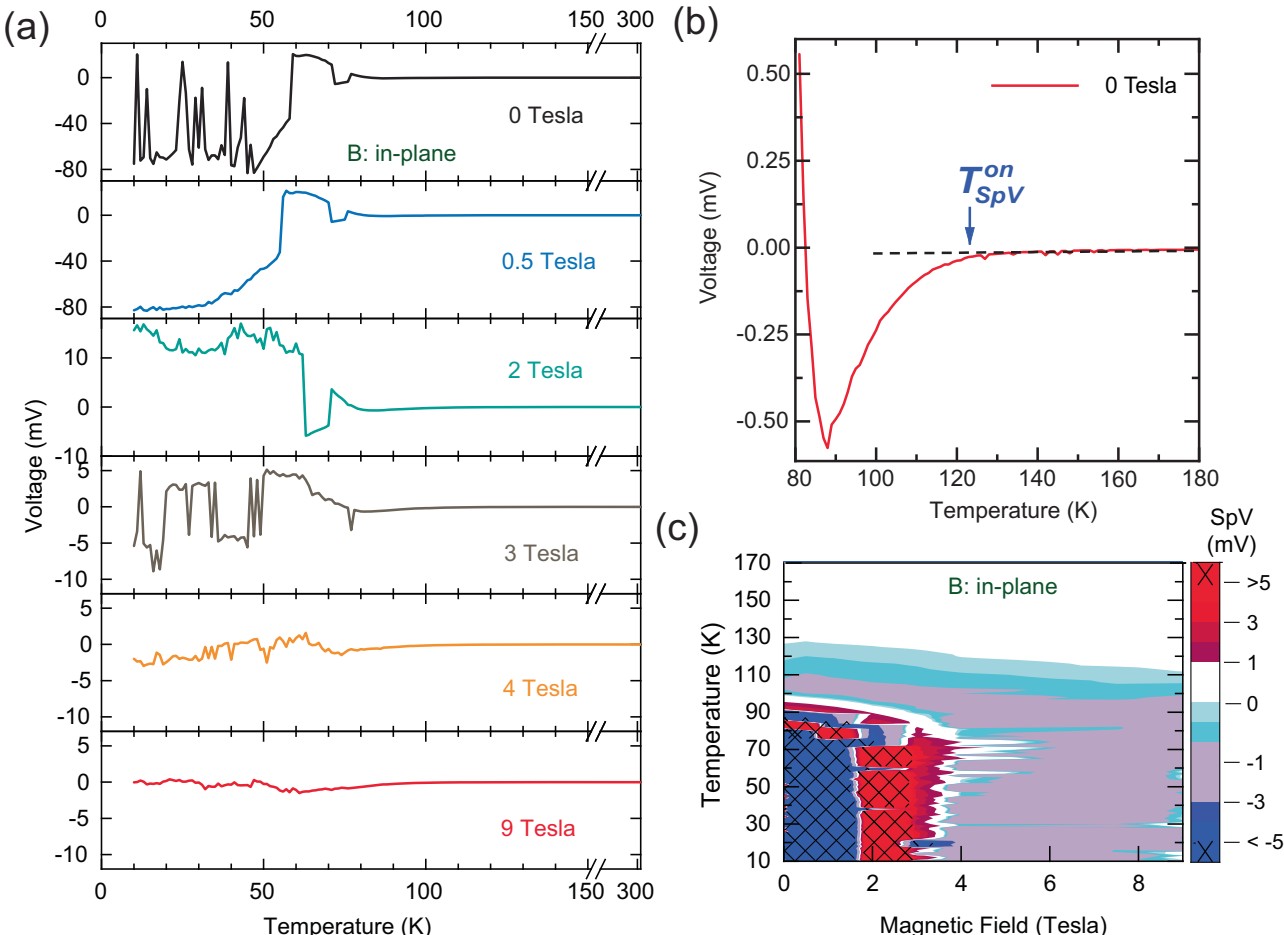

**Fig. 2 | SpV signal at zero applied current. a** Temperature and (in-plane) magnetic field dependence of the SpV between the inner contacts that develops during cooling without a current applied between the outer contacts (see Fig. 1a). **b** Magnified view of the gradual development of the SpV at zero magnetic field with an onset temperature of about 120 K. **c** Contour plot of the SpV versus temperature and magnetic field.

YBCO layers. Note that the latter is only weakly affected by an in-plane magnetic field of 9 T. Figure 1d displays the M-T curves in field-cooled mode that have already been reported in ref. 18. At low fields of 50 Oe and 100 Oe they confirm that the NCSMO layers have a weak FM moment of about 0.1–0.2 $\mu_B$/Mn with a transition temperature of $T^{Curie} \approx 110$–120 K (as marked by an arrow in the inset). Towards higher fields, the M-T curves reveal that the FM moment gets strongly enhanced and eventually saturates around 1.7 $\mu_B$/Mn while its onset temperature increases to $T^{Curie} \approx 250$ K.

### Temperature dependence of the SpV

Figure 2 shows the temperature and magnetic field dependence of the SpV that develops between the inner line contacts while the outer ones remain open. At zero magnetic field, the SpV signal develops gradually below an onset temperature of about 110–120 K, as detailed in the magnified view shown in Fig. 2b. Upon approaching the SC transition at $T_c^{ons} \approx 80$ K, the SpV suddenly increases in magnitude and eventually reaches remarkably high values of up to 80 mV. Moreover, the SpV signal starts to exhibit sudden, step-like changes, some of which give rise to a sign reversal.

### Magnetic field dependence of the SpV

The data in Fig. 2 show that the SpV signal is overall strongly suppressed by a large magnetic field. In particular, the contour plot in Fig. 2c reveals that the SpV undergoes some major changes around 2 and 4 T where the magnitude and even the sign change rather suddenly.

Figure 3a–f shows corresponding magnetic field loops of the SpV at several representative temperatures. Most remarkable are the loops at 50 K and 80 K in panels (c) and (d) which show strong SpV signals that exhibit large, jump-like changes of the magnitude and even the sign in the range of 2–2.5 T. Except for some deviation of the virgin curve at 50 K, these loops are reversible and exhibit a hysteretic behavior below about 4 T with coercive fields of about 0.5–0.8 T. Panel (g) shows a series of partial field loops at 70 K which confirm that the SpV signal exhibits a memory effect and thus can be repeatedly switched in successive loops as well as in partial loops. The magnetic field loops at 10 K and 30 K show large differences between the virgin curve and the subsequent field-loops. In particular, the SpV at 10 K shows an irreversible response that is indicative of a frozen or glassy behavior. Finally, in the field-loops at 90 K and 100 K, the SpV signal is strongly reduced in magnitude and evolves rather continuously without a clear hysteresis or irreversible changes, except for some random fluctuations that are well above the typical noise level. These trends indicate that the mechanism underlying the SpV signal involves a domain state which at elevated temperature can be readily and reversibly modified by the magnetic field, whereas it starts to freeze and eventually exhibits a glassy behavior at low temperature.

Figure 4 shows a comparison of the magnetic field loops of the voltage signal (between the inner contacts) taken (a) at zero current and (b) at a finite current of $I = 10$ μA applied across the outer contacts. It highlights that a large current can reduce the magnitude of the SpV signal and suppress the large, jump-like changes and the related hysteresis and memory effects. We will further discuss below how these

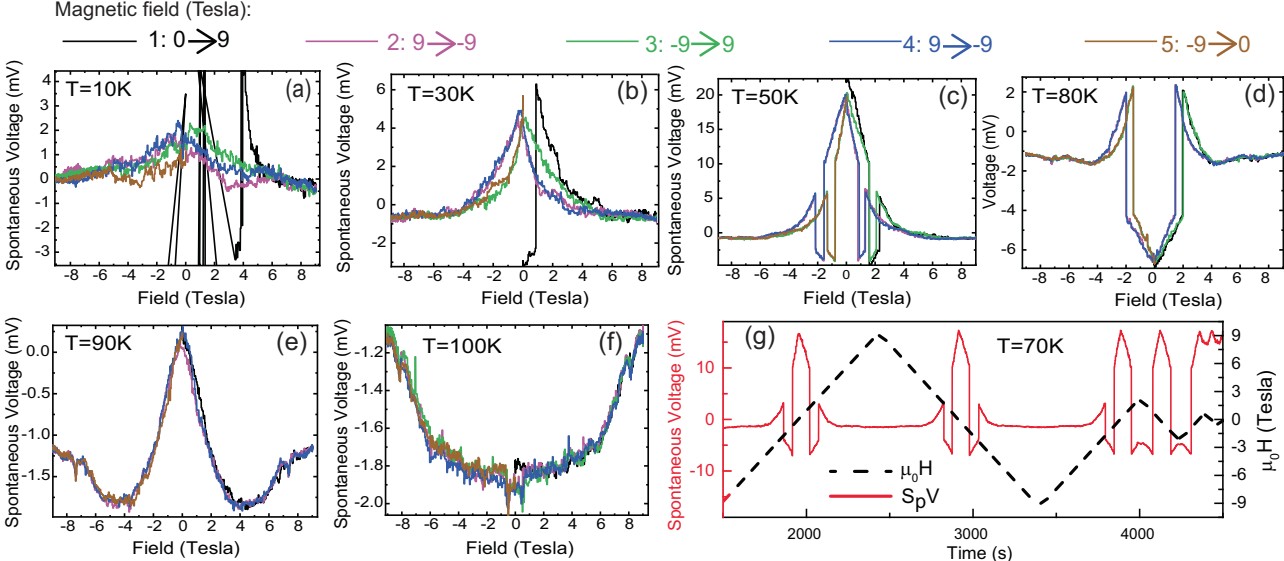

**Fig. 3 | Magnetic field loops of the SpV signal showing hysteresis and memory effects. a–f** Magnetic-field-loops of the SpV at temperatures between 10 K and 100 K. The order of the successive field sweeps is shown by the color code that is indicated on top. In particular, the loops at 50 K and 80 K reveal a switching behavior with sudden changes of the magnitude and even the sign of the SpV that

exhibit a hysteresis and a strong memory effect. **g** Demonstration of the memory effect of the SpV during full and partial H-field loops at 70 K. Solid lines denote the value of the SpV (left y-axis) and dotted lines the magnetic field (right y-axis). The colors in (**a–g**) follow the same code as indicated above the Figure.

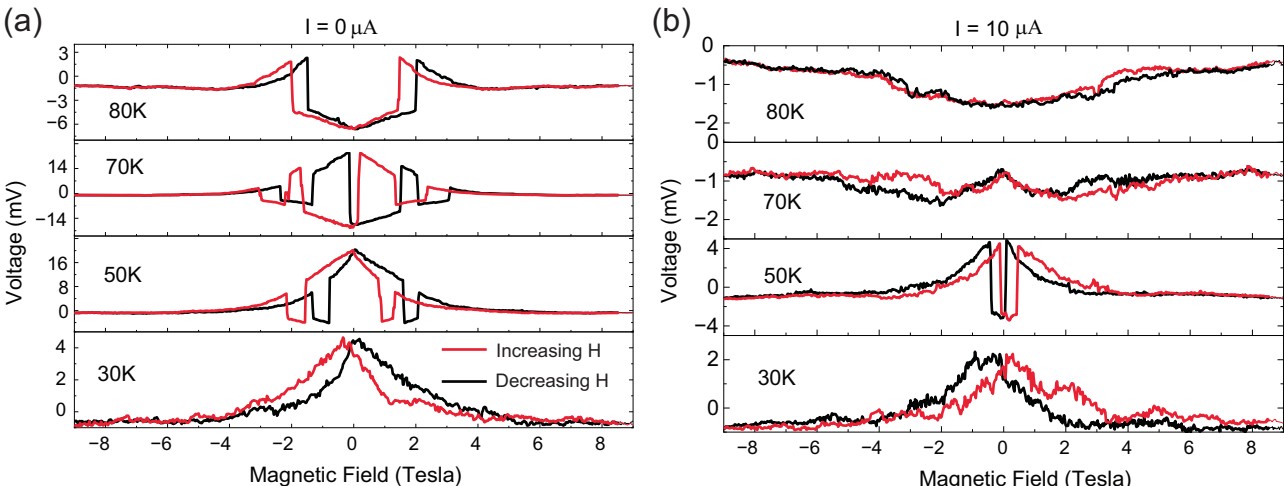

**Fig. 4 | Effect of an applied current on the magnetic field loops of the SpV.** Magnetic field loops (without virgin curves) of the SpV signal of the N10Y7-SL for applied currents of (**a**) zero and (**b**) 10 μA. The colors in (**a**) and (**b**) follow the same code as indicated in the legend of (**a**).

current- and magnetic-field-induced changes of the SpV signal can be understood in terms of the competition between nearly degenerate phases that are either conducting, non-polar and ferromagnetic (FM) or insulating, polar and antiferromagnetic charge/orbital ordered (AF-COO).

## SpV for wire-bonded contacts

Figure 5a shows that the SpV signal also develops when using wire-bonded contacts which cover a much smaller area (with a diameter of about 50 μm) and yield a direct connection to the buried YBCO layers. As compared to the line-contacts in Fig. 2a, the magnitude of the SpV is reduced by about an order of magnitude and the curves do not show pronounced jump-like changes as for the line contacts. Nevertheless, the successive cooling and warming sweeps

reveal a clear SpV signal that starts to develop below about 120 K, increases in magnitude towards low temperature, and exhibits a memory effect that can be erased by heating the sample well above the SpV onset temperature.

The reduced magnitude and the absence of large, jump-like changes in the SpV signal of the wire-bonded contacts can be understood in terms of their direct connection with the YBCO layers, which helps to reduce the charging effects from the topmost insulating NCSMO layer. Moreover, the extended line contacts may act as antennas which enhance the coupling to the electronic noise for which the maximal intensity likely occurs in the microwave region[1]. Figure 5b displays the time evolution of the SpV signal for the wire-bonded contacts which confirms that this phenomenon persists as long as the sample is kept at the designated temperature.

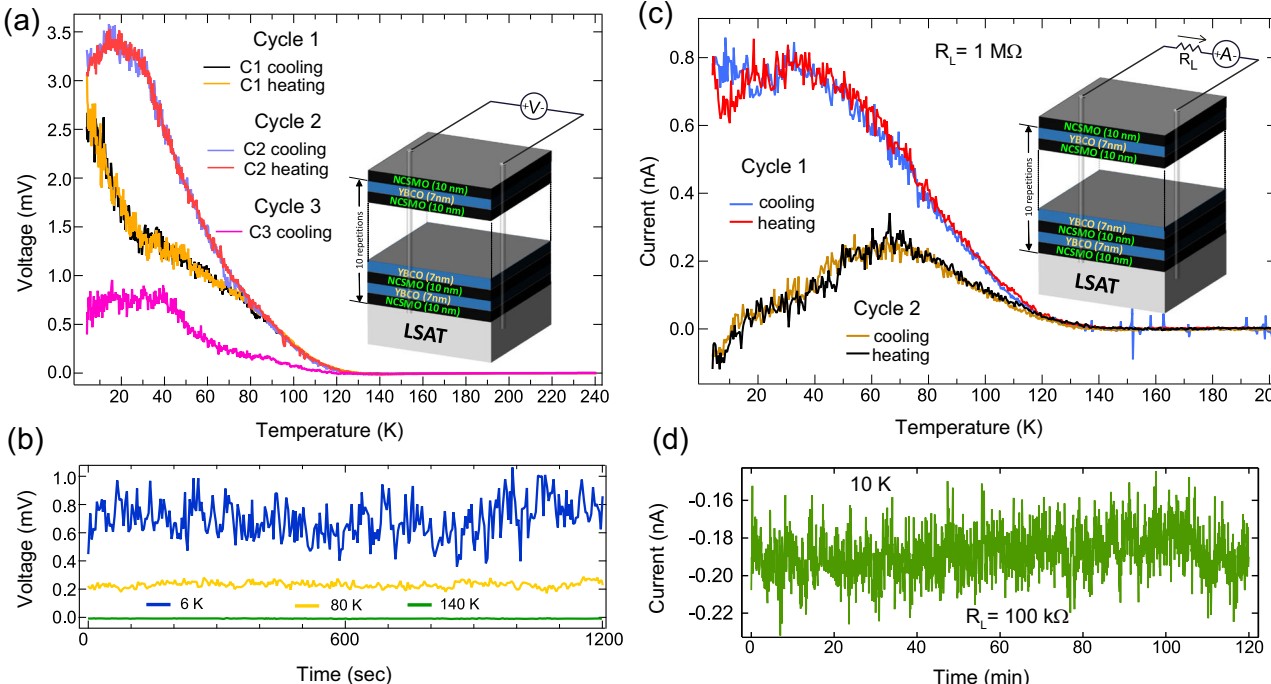

**Fig. 5 | Spontaneous voltage (SpV) and related current (SpC) for a N10Y7-SL with wire-bonded contacts. a** Temperature sweeps of the spontaneous voltage (SpV) between wire bonded contacts showing a memory effect of the SpV that is erased after warming to 250 K. Inset: Sketch of the wire-bonded sample and the circuit to probe the spontaneous voltage. **b** Time dependence of the SpV at fixed temperatures of 6 K, 80 K and 140 K, showing the persistence of the SpV signal below about 120 K. Note the enhanced noise at 6 K. **c** Temperature dependence of the spontaneous current (SpC) that the wire-bonded sample generates across an external resistance ($R_L$) of 1 MΩ when no external voltage is applied. Inset: Sketch of the wire-bonded sample and the circuit to probe the spontaneous current. **d** Time dependence of the SpC current across an external resistor ($R_L$ = 100 kΩ) at 10 K.

## Persistent current (SpC) driven by the SpV

Figure 5c demonstrates that the SpV can be used to generate a spontaneous current (SpC) across an external circuit consisting of a series of an ohmic resistance of $R_L$ = 1 MΩ and a Keithley/Tektronix 6487 Picoammeter. In analogy to the SpV curves in Fig. 5a, the SpC through the external circuit gradually appears below about 120 K and increases in magnitude toward low temperature, where it reaches several hundred picoamperes. Like the SpV in Fig. 5a, the SpC in Fig. 5c exhibits a clear memory effect upon cooling and warming that can be erased if the sample is heated well above the transition temperatures of the FM and AF-COO. Finally, Fig. 5d confirms that at a constant temperature of T = 10 K the SpC persists on the time scale of hours where it shows no sign of a decrease.

## SpV versus YBCO layer thickness

Finally, Fig. 6 displays the evolution of the magnitude of the SpV signal as a function of the YBCO layer thickness, $d$, for a series of NCSMO/YBCO/NCSMO trilayers. Shown are the maximal values of the SpV signal at zero applied current and zero magnetic field. The SpV signal exhibits an approximately exponential decrease as a function of $d$. This trend is consistent with the scenario that the SpV signal has its origin in the charged domain walls of the NCSMO layers which create electric fields that emanate into the YBCO layers where they get partially screened. A fit with the function $SpV_{max} \sim A \cdot \exp(-d/2l^{sc})$, where $A$ is a constant, yields a length scale for the screening of $l^{sc} = 0.5 * ln\left(\frac{0.001}{A}\right) \approx 4.7$ nm, which in the following we denote as the screening length. The factor of two in the above equation accounts for the circumstance that the YBCO layer is interfaced with the top and bottom NCMSO layers.

## Discussion

In the following we discuss a minimal model to account for our key observations: (i) the formation of a large SpV that (ii) can drive a

persistent electric current across an external circuit, that (iii) develops gradually below about 120 K, more or less in parallel with the FM order (at $B \approx 0$), that (iv) increases in magnitude towards low temperature, that (v) undergoes sudden changes of the magnitude and even the sign during temperature-sweeps and magnetic field ($H$) loops, that (vi) exists in zero magnetic field with even symmetry under time reversal (magnetic field), and that (vii) exhibits clear hysteresis and memory effects during field sweeps.

At first, we emphasize that the above-described features are incompatible with a thermoelectric effect that arises from a thermal gradient across the YBCO/NCSMO heterostructure or the wires connected to it. Such a scenario could not explain the persistence of the SpV signal and the related currents, since the sample is glued to the cold finger of the cryostat and thus strongly coupled to a heat bath. Accordingly, the thermal gradient would depend on the cooling rate and decrease over time while the sample is dwelling at a fixed temperature. Moreover, the typical magnitude of such a thermoelectric voltage is on the order of microvolts, rather than millivolts, and it has a maximum at elevated temperatures and decreases strongly towards low temperature (see Supplementary Note 2). Finally, a thermoelectric voltage is not expected to undergo jump-like changes during temperature and magnetic field loops nor to exhibit a memory effect. For the reasons described above, we can exclude any alternative mechanism for which the energy that is required to develop a SpV and drive a persistent electric current through an external circuit is drawn from an external heat bath. Here the ubiquitous decrease of the lattice contribution to the thermal conductivity and the heat capacity would cause a strong reduction of the energy transfer at low temperature, and as a consequence also of the SpV.

In the following, we argue instead that a rectification of electronic noise based on an asymmetric, ratchet-like electronic potential is at the heart of the SpV and the persistent current effects that require a

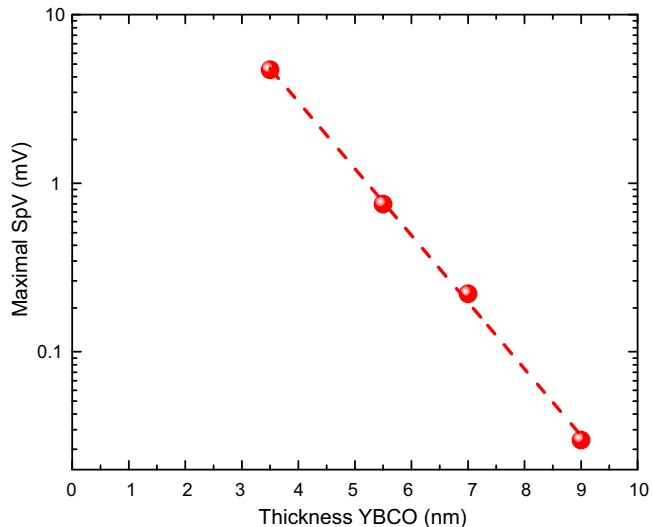

**Fig. 6 | SpV versus YBCO layer thickness.** Variation of the maximal SpV signal as a function of the YBCO layer thickness, $d$, for NCSMO(10 nm)/YBCO(d)/NCSMO(10 nm) trilayers. The dashed line is a linear fit to the plot.

steady-state generation of power. Here, the device acts as the receiver and the source of energy is provided by the environmental electronic fluctuations/noise that the sample is exposed to inside the magneto-cryostat of the physical property measurement system (PPMS). As detailed in the Supplementary Note 3, we have verified that the SpV signal is sensitive to the change of the driving mode of the PPMS magnet, as well as to electromagnetic radiation created with an a.c. current generator which is connected to a coil that is located inside the PPMS sample chamber (Supplementary Note 4). Moreover, in corresponding a.c. transport measurements, we detected a $2^{nd}$ harmonic signal that develops in parallel with the SpV below about 120 K and increases in magnitude towards low temperature (see Supplementary Note 5).

We argue that the ratchet-type electronic potential has its origin in the complex domain state of the NCSMO layers. The manganites are indeed known for their rich spectrum of magnetic and electronic orders that results from the competition between the superexchange, double exchange and Jahn-Teller interactions. A famous example is the colossal magnetoresistance (CMR) effect which occurs e.g. in $La_{1-x}(Ca,Sr)_xMnO_3$ with $0.2 < x < 0.5$, for which an external magnetic field can tip the balance in favor of the double-exchange and thus induce a transition from a paramagnetic, insulator-like to a ferro-magnetic, metallic state[19–21]. Equally intriguing phase competition phenomena occur for the systems with more strongly tilted Mn-O-Mn bonds, e.g. in $(Pr,Nd)_{1-x}(Ca,Sr)_xMnO_3$ with $0.3 < x < 0.5$, for which the Jahn-Teller interaction is enhanced. A phase segregated state develops here that is composed of so-called Zener-polarons[22] that are either long-range ordered, and give rise to a phase with charge-exchange-type AF and charge/orbital order (AF-COO), or they are in a disordered "liquid-like" state (D-phase).

Notably, the distorted Mn-O-Mn bonds in the AF-COO state give rise to a polar electric moment and a subsequent domain state with charged domain walls[22] (See Supplementary Note 1). The reported size of these polar domains is about 20 nm, the polar moments are directed along the $a$-axis and cause head-to-head or tail-to-tail domain walls whose normal vectors are along the polar moments[22]. The estimated values of the net polarization and the dielectric constant of $P \approx 40$ mC/m$^2$ and $\varepsilon \approx 35$, respectively, yield a sizeable change of the electric potential of ~2 V between the opposite types of domain walls and an electric field strength of $E \approx 10^8$ V/m. The corresponding surface charge density of the domain walls amounts to about

0.07 electrons per pseudocubic u.c. of NCSMO (see Supplementary Note 4 for more details).

The transition to this charge/orbital ordered and polar state occurs around $T_{COO} \approx 230$–$250$ K, depending on the composition and doping level[21,23]. It appears to be of an order-disorder type with a gradually increasing volume fraction of the COO phase towards low temperature[22]. The magnetic transition of the AF phase occurs around $T^N \approx 140$–$170$ K[24]. An additional FM transition occurs in the range of $T^{Curie} \approx 110$–$130$ K[20] in zero or low magnetic fields. Here the magnetic moment and thus the volume fraction of this FM phase amounts to less than 10% of that in the bulk FM state of e.g. $La_{2/3}Sr_{1/3}MnO_3$ with a saturation value of about 3.7 $\mu_B$/Mn[21]. Whether this FM minority phase develops merely at the expense of the AF-COO phase or also of the D-phase is unknown. There exists, however, clear evidence of a strong competition between the FM and the AF-COO phases. For example, the FM moment and the related volume fraction can be strongly enhanced with a large external field that tips the balance between the competing AF-COO and FM phases towards the latter. Similar changes can be induced with other external stimuli, like electric currents[22] or high energy photons[25,26].

In the following, we propose a minimal model to account for the SpV and its unusual temperature and magnetic field dependence, that is based on a ratchet-type potential that arises from the competition between the polar COO phase and the non-polar FM phase and their pinning-induced inhomogeneous lateral distribution. Figure 7a shows a sketch of the electronic potential for a balanced polar domain con-figuration with a vanishing total polar moment ($\langle p \rangle$=0) that minimizes the energy due to electric stray fields and occurs at elevated tem-peratures where the FM phase is absent, i.e. at $T > T^{Curie}$. For simplicity, the sketch does not show the coexisting non-polar D-phase, which is likely segregated from the polar COO phase, see e.g. ref. 19. The sketch in Fig. 7b shows how the electric potential of this compensated polar domain state is modified when the nonpolar FM phase appears at $T < T^{Curie}$ and starts to compete with the polar AF-COO phase. Here we assume that the nonpolar FM phase develops right at the head-to-head domain walls for which the electric potential is maximal and thus attractive for electrons. Such a scenario is suggested by the doping phase diagram of the manganites where the FM phase (AF-COO phase) prevails at lower (higher) hole doping[20,21]. Figure 7b shows that the non-polar FM phase gives rise to a reduction of the electronic potential in its vicinity. It also reveals that a spatially inhomogeneous distribu-tion of these non-polar FM domains causes an asymmetry of the electronic potential and subsequently a SpV between contacts that cover regions with a different density of the FM patches (note that the sketch is not to scale with respect to the size of the contacts and the polar domains). Such an inhomogeneous distribution of the FM phase is likely caused by a pinning to local or extended defects that exhibit a density gradient across the sample. Such a lateral gradient may arise e.g. from a terrace structure due to an imperfect surface cut of the substrate[27], from a long-ranged strain gradient across the sample due to the lattice mismatch between the substrate and the thin film[28], or from a lateral variation of the chemical composition of the thin film that is inherent to the PLD growth process, due to the complexity of the ionic composition and dynamics of the plasma plume[29].

The occurrence of a large SpV signal furthermore requires that the volume fraction of the non-polar FM- and D-phases remains below the percolation limit, above which conducting pathways develop across the multilayer that can short the contacts. It is equally important that the YBCO layers are rather thin such that they can only partially screen the electric fields that emanate from the charged domain walls. The charge carriers of the YBCO layers are thus localized in the vicinity of the polar COO phase and remain mobile only in the regions of the non-polar FM- and D-phases. A percolation effect due to a growing fraction of the nonpolar D-phase, in combination with the suppression of the FM order above $T^{Curie}$, therefore can explain the strong reduction

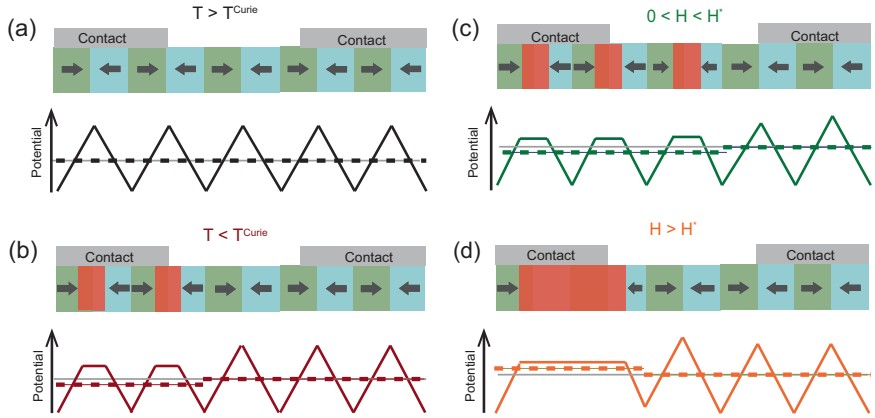

**Fig. 7 | Illustration of the origin of the SpV from competing polar AF-COO and non-polar FM phases. a** Electronic potential for a compensated polar domain state ($\langle p \rangle = 0$) of the bare AF-COO phase at $T^{Curie} < T < T_{COO}$. The electronic potential, $\Phi$, and its average over pairs of neighboring polar domains, $\langle \Phi \rangle$, are shown by solid and dashed lines, respectively. The green and cyan regions indicate regions with opposite polarity, where the arrows correspond to their polar moment. **b** Reduction of $\Phi$ in regions where the non-polar FM domains (red coloring) develop at the polar head-to-head boundaries. **c** Sudden change of the electronic potential upon increasing H-field. Below the threshold field $H < H^*$ the non-polar FM domains develop around individual head-to-head boundaries where they reduce $\Phi$. **d** Above $H^*$ the FM domains start to cluster and cover extended regions giving rise to a sudden increase of $\langle \Phi \rangle$.

of the SpV signal above 120 K. Likewise, the reduction of the SpV signal by a large magnetic field (at $T \ll T^{Curie}$) can be readily understood in terms of an enhanced volume fraction of the nonpolar FM phase.

Next, we show that the above-described minimal model may also account for the sudden changes of the magnitude and even the sign of the SpV that occur in the experimental field loops in Fig. 3c, d. The sketches in Fig. 7c, d show that a sudden change of the electronic potential landscape $\Phi$ can occur as the volume fraction of the nonpolar FM regions increases and they eventually start to merge forming extended clusters. Note that above a certain threshold field value $H^*$ such a clustering can be energetically favorable, since it enhances the electronic potential in the vicinity of the FM phase (as compared to the case where the FM patches are confined to single domain walls). Moreover, the FM clusters might be forming extended conglomerates for which a switching between different ordering patterns can be induced with a magnetic field. The memory effect of the SpV signal can arise here because the various ordering patterns are preset by the spatial distribution of the underlying pinning centers (due to the defects discussed above). Such a scenario also seems consistent with the finding that the large and sudden changes of the SpV signal occur only at elevated temperatures, i.e. between about 40 K and 80 K, where the thermal energy helps to overcome the energy barriers between the various energetically favorable patterns and the SpV signal thus becomes reversible.

The minimal model shown in Fig. 7 is also compatible with the almost symmetric shape of the magnetic field loops of the SpV signal for which the magnitude and sign are independent of the direction of the applied field. Here, the magnetic field changes the energy balance between the competing AF-COO and FM phases by an amount that is proportional to the magnetization squared[30] and thus independent of the sign of the field.

Finally, we further discuss the role played by the YBCO layers, which enable the lateral charge flow and thus are essential for using the SpV to drive a persistent current across an external circuit. These YBCO layers are fully oxygenated and thus close to optimal hole doping. Their superconducting (SC) transition temperatures of $T_c \approx 70-80$ K are indeed only moderately reduced as compared to the bulk value of $T_c \approx 90$ K[31,32]. However, despite a robust SC pairing strength, the macroscopic SC phase coherence of these YBCO layers appears to be very fragile, similar to granular superconductors[16,33].

A finite SpV signal only occurs as long as the thickness of the YBCO layer remains below the screening length, $l^{sc}$, (introduced in Fig. 6) such that it can only partially screen the electric field that emerges

from the charged boundaries of the polar COO domains in the neighboring NCSMO layers. Moreover, it is mandatory that the NCSMO layers contain a sufficiently low fraction of the nonpolar FM or D-phases as to avoid the formation of conducting pathways across the YBCO layers. The charge carriers of the YBCO layers are thus weakly localized. However, they can be activated by absorbing photons from the background radiation, which enables them to overcome the barriers of the partially screened ratchet-type potential and develop a lateral charge flow. This interpretation agrees with our finding that a corresponding persistent current effect does not occur for a single NCSMO layer, as shown in the Supplementary Note 6. It is also confirmed by the variation of the SpV signal as a function of the thickness of the YBCO layer shown in Fig. 6 which exhibits the expected exponential trend, $SpV_{max} \sim A \cdot \exp(-d/2l^{sc})$ with $l^{sc} \approx 4.7$ nm. This value is quite a bit larger than the estimated Thomas-Fermi-screening length of about 0.1 nm[34] (given the carrier density of optimal doped YBCO[35] of $\sim 10^{21}$ holes/cm³). The enhanced screening length may result from the quasi-two-dimensional structure of YBCO, even though, corresponding Thomas-Fermi-theory based calculations suggest that a particle with an elementary charge $e^-$ is almost fully screened by a single CuO₂ plane[36]. Nevertheless, in these calculations it is not considered that the high-$T_c$ cuprates are hole doped Mott-insulators which have a rather limited capacity for screening positive charges. This is especially true for the interfacial CuO₂ layers for which the hole doping is reduced by a charge transfer across the interface[37] as well as by the absence of a CuO chain layer (and thus charge reservoir) right at the interface[38,39].

To summarize, our study has identified YBCO/NCSMO multilayers as promising artificial materials for creating devices for efficient energy harvesting of electronic background noise. In particular, we have shown that they can develop rather large SpV signals on the order of tens of millivolts in a wide range of temperatures and magnetic fields. We have also demonstrated that this SpV can drive a persistent current through an external circuit and thus perform electrical work. Moreover, we have found that the magnetic field loops of the SpV signal exhibit pronounced hysteresis and memory effects which can be readily modified via the temperature and applied electric currents, and most likely also electromagnetic radiation[40–42]. These YBCO/NCSMO multilayers thus provide opportunities for developing self-powered multifunctional devices and non-volatile memories.

We emphasize that our results are distinguished from previous work[43–46] on zero-field non-reciprocal transport since we observe large SpV and SpC effects over a temperature range of more than 100 K.

Moreover, a SpV signal with even-in-field symmetry, to our best knowledge, was only reported for disordered Nb channels and thin films[47] where a SpV at zero-field appears only in a narrow temperature window around $T_c \sim 3$ K.

Our study is also of great interest from a fundamental science perspective. In the first place, it has highlighted the complexity of the domain state of the NCSMO layers which in addition to various orbital and magnetic orders give rise to a polar order of the COO phase and a subsequent charging of domain walls, which has so far received limited attention. Our study has also revealed an important role of the YBCO layers which are nominally metallic and superconducting. In particular, it has provided evidence for a rather large length scale ($l^{sc} \approx 4.7$ nm) of the screening of the electric fields that emanate from the charged domain boundaries of the neighboring NCSMO layers. This finding has direct implications for electric field-effect studies in related heterostructures with thin superconducting layers of high-$T_c$ cuprates, for which the direct interaction of the static electric field with the superconducting order parameter has been widely neglected but is recently obtaining increased attention[48–54].

Our findings also indicate a possible correlation between the onset below 80 K of the superconductivity in the YBCO layers and the strong enhancement and large fluctuations of the SpV. Such a coincidence could imply a back-action of the superconducting order in YBCO on the domain state of the neighboring NCSMO layers. Such a cross-talk might be mediated by the interaction of the vortices of the SC with the FM domains. Likewise, it could be related to the poor screening capability of the YBCO for positive charges (due to the above discussed proximity to a Mott-insulator state) and the subsequent formation of a pattern of alternating SC and insulating domains in the YBCO layer. Such a domain state could act like a network of Josephson-type junctions and thus contribute to the sudden changes of the magnetic field loops in Fig. 3 and their pronounced memory effects.

Microscopic scanning probe studies of the structure of the complex domain state of these YBCO/NCSMO heterostructures will be required in order to fully understand the mechanism underlying their SpV and SpC effects. It will also be of great interest to explore whether defect or strain engineering can be used to further enhance the amplitude and control the sign of the SpV and SpC signals and to tailor the pronounced hysteresis and switching effects in the temperature and $H$-field loops. Likewise, it remains to be further explored how the SpV and SpC signals depend on the polarization, spectral distribution and intensity of the electromagnetic background radiation, as well on the size and shape of the electric contacts of the samples.

## Methods

### Sample preparation and characterization
The growth and characterization of the used NYN superlattice has been detailed in sec. II.A of ref. 18. The measurements of spontaneous voltage and current generation was also reproduced for YBCO (10 nm)/Nd$_{0.65}$(Ca,Sr)$_{0.35}$MnO$_3$ (7 nm)/YBCO (10 nm) trilayers, which were prepared following the same growth recipe.

### Measurement of electrical properties
We have used a Keithley 2182 A nanovoltmeter and a Keithley DMM6500 digital multimeter to measure the voltages, while Keithley/Tektronix 6487 picoammeter was used to measure current. A Keithley 6200 was used as current source for constant-current measurements. All electrical measurements were performed in QD PPMS cryostats, which was interfaced with the above-mentioned devices.

## Data availability
The source data that support the findings of this study are available in figshare with the identifier(s): https://doi.org/10.6084/m9.figshare.28624961.

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

## Acknowledgements

The work at the University of Fribourg was supported by the Swiss National Science Foundation (SNSF) through Grants No. 200020_172611 and 200021_212050. Yu. G. Pashkevich acknowledges the financial support of the Swiss National Science Foundation through the individual grants IZSEZ0_212006 and IZSEZ0_215824 of the «Scholars at Risk» Program. R.C acknowledges the support by a grant of the Ministry of Research, Innovation and Digitization, CNCS-UEFISCDI, project number PN-III-P1-1.1-PD-2021-0238, within PNCDI III and a scholarship "Cercetare postdoctorală avansată" funded by the West University of Timisoara, Romania. D.M. acknowledges support by the project Quantum materials for applications in sustainable technologies, Grant No. CZ.02.01.01/00/22_008/0004572. The authors would like to thank Dr. Marsik Premysl for his help in assembling the external noise inducer setup. The authors would also like to thank the mechanical and electronic workshops at the Physics department of the University of Fribourg for their help in manufacturing different components and regular maintenance of the equipment.

## Author contributions

The samples were grown by J.K. and M.S. The characterization was performed by S.S. and M.S. Measurement of spontaneous voltage by line-contacts was performed by M.S. Measurement of spontaneous voltage and current in wire-bonded samples were performed by Sh.S., M.M., S.S. and L.D. The magnetic properties were measured by M.S., R.C. and S.S. The measurement of the second harmonic component in a.c. transport and rectification of externally sourced noise signal was done by S.S. and R.T. The analysis of the data was done by M.S., S.S. and C.B. S.S., C.B., Y.P. and D.M. contributed in the theoretical understanding and modeling. Finally, C.B., S.S., Sh.S., Y.P. contributed in the writing of the manuscript and R.C. helped in formatting and processing of Figures. S.S. stands for Subhrangsu Sarkar and Sh.S stands for Shamashis Sengupta.

## Competing interests

The authors declare no competing interest.
