## [Transparent Peer Review file · Nature Communications]

Spontaneous voltage and persistent electric current from rectification of electronic noise in cuprate/manganite heterostructures

Corresponding Author: Dr Subhrangsu Sarkar

Version 0:

Reviewer comments:

Reviewer #1

(Remarks to the Author)

In this work, the authors report the appearance of a spontaneous voltage without applied current in YBCO/NCSMO multilayers. They found that this voltage can be heavily affected by magnetic field and temperature. They propose a possible explanation for this effect based on electronic noise rectification and the nucleation of non-polar FM domains below the Curie temperature of the material, competing with polar COO domains. I have the following remarks concerning this work:

- 1) In figure 1d it would be necessary to add the magnetization curve for 0.5T, as it is one of the fields used in figure 2a.
- 2) The magnetic field is applied in plane in Figure 1, and I suppose the direction is the same for figure 2, but it could be better to mention it in the text.
- 3) The temperature at which the SpV starts is around 120K, as shown in figure 2b, and appears to be negative and of the order of 0.5mV. Nevertheless, the larger effect, which can reach -80mV starts to appear below the T_c of YBCO, around 80K (which can also be seen in figure 2b). To me the SpV appears to have its origin in the NCSMO, but it is heavily increased when there are superconducting layers. Have the authors tried to measure different samples changing either the NCSMO for other magnetic material or the YBCO for other non superconducting material? For example, multilayers of LSMO/YBCO or multilayers of NCSMO/LaNiO₃. This could clearly point towards the main contribution to the effect, either the magnetic (and polar) properties of the NCSMO or the superconducting state.
- 4) The authors comment about figure 2c are: "Fig. 2(c) reveals that this suppression of the SpV does not happen gradually but rather occurs in two major steps around fields of about 2 and 4 Tesla. At the first step slightly below 2 Tesla the SpV signal is reduced by about an order of magnitude." From figure 2c what I observe is that there is a change in the sign of the SpV below 2T, not an order of magnitude decrease. I also observe a gradual decrease of the SpV from 3 to 4T (goes from more than 5mV to -1mV). Maybe it is a problem of the scale (only between 5 and -5 mV) or maybe it is a problem of the colours (the same ones for -1 to -3 mV), but I do not agree with the author's comment about the figure.
- 5) In figure 3, the maximum value of the SpV is 20mV at 50K. Nevertheless, looking at figure 2, there should be maximum (absolute) values of up to 80mV. Moreover, I would expect that the maximum change in SpV in function of field would be around 20-30K, where the voltage changes from -80mV to 0mV according to figure 2. I understand that in one case the V vs T curves are measured with continuously applied field, while the V vs H not, but why is the behavior so different between figure 2 and 3?
- 6) In figure 4, could it be possible to present also the curves for -10uA? I do not understand how in the case of T=50K, for example, the voltage of the multilayer is lower when applying 10uA than with 0uA. In principle, and looking at figure 1c, all the current should flow through the YBCO layers, giving a zero resistance state. How can the current flowing through the YBCO affect the SpV effect in the NCSMO layers? It could be interesting to show current-voltage curves to have a more general view of the effect of applied currents. Also, when the authors say they apply 0uA, it is with the current source on, applying 0uA or with the current source off?
- 7) Figure 6 shows the maximum amplitude of the SpV in function of YBCO thickness. Following the trend, have the authors tried to use a YBCO thin enough (1 or 2 unit cells) so that the YBCO is insulating? What would the authors expect to happen then?
- 8) With respect to the SpV originating from environmental electronic noise, could the authors purposely introduce noise, for example with an RF antenna, to controllably increase the SpV effect?
- 9) With respect to the discussion, how do the authors explain that, for example, for the SpV measured at 2T, the SpV starts to

appear at 120K while magnetization measurements show that the system is magnetic at temperatures as high as 225K? According to the discussion, the presence of magnetic domains (and hence, the absence of polar domains) is responsible for the SpV

Small changes:

- In line 70 CE-type AF is not defined (Charge-Exchange-type)
- The references 18 to 21, which the authors cite in the introduction to talk about $(\text{Pr,Nd})_{1-x}(\text{Ca,Sr})_x\text{MnO}_3$, only refer to $\text{Pr}_{1-x}(\text{Ca,Sr})_x\text{MnO}_3$ and other manganites, but not to $\text{Nd}_{1-x}(\text{Ca,Sr})_x\text{MnO}_3$, which is the material used in this work. They should cite bibliography about this specific material, or at least a manganite with Nd.
- In figure 3 the color code should also be written in the figure text to make it clearer. Also in figure 3f, the minus sign of the SpV cannot be seen properly.
- Lines 176 to 178 indicate that the scenario of a competition between different phases in NCSMO has been described already, but it is explained later in the paper.

In summary, the work presented here is original, scientifically valid (although the origin of the effect may be subject to doubt). It is an interesting work, but I think its relevance can be limited to scientists working outside of the complex oxides field, as the properties needed for this effect can only be found in this type of materials. Therefore, I would like my remarks to be addressed before I can recommend it for publication in Nature Communications.

Reviewer #2

(Remarks to the Author)

In this article, Soulier et al. stacked an artificial cuprate/manganite-stacked superlattice structure, and investigated the electronic transport properties of such a structure. Apart from the expected superconducting transition of YBCO and magnetic transitions of NCSMO layers, it is quite interesting that a spontaneous voltage signal below ~ 130 K was observed, with an amplitude being as large as tens of mV. Such spontaneous voltage signal can be modulated by external magnetic field due to the modulation of domain structure of NCSMO layers, and shows potentials for memory functionality. Generally, I think this work is interesting and innovative enough to be published in Nature Communications. But still, there are some points that I think the author should consider prior to the acceptance of this work.

1. I found that the Introduction section is kind of too long to catch the main parts that the authors intended to introduce. Therefore, I suggest the authors shorten their Introduction section and highlight the key points.

2. The heterostructure is similar to the Ba-Nb-S system where the superconducting layer is sandwiched by non-superconducting materials (Science 370, 231 (2020)) where 2D superconductivity and FFLO state was claimed. Similarly, can the authors observe any feature of superconducting YBCO in such a heterojunction that is different from the bulk YBCO samples?

3. The orientation of magnetic field (in-plane or out-of-plane) should be clarified in Fig. 2. A schematic illustration is suggested.

4. Are such observations of SpV expected to be reproduced based on FM materials with out of-plane magnetic easy axis? For instance, the insulating FM material, $\text{Cr}_2\text{Ge}_2\text{Te}_6$, can have multi-domain structure with net magnetization ~ 0 (similar to the illustration in Fig. 7 in this manuscript).

5. I am quite curious the role Josephson effect plays in such an SpV effect, since the superconducting order parameter between adjacent YBCO nanoflakes should form a weak-coupling Josephson junction. Since 0-junctions and/or pi-junctions can form in SC/FM/SC heterostructures, I am wondering if these junctions exist in such a heterostructure. Will they affect the electronic properties of the heterojunction?

6. The authors used " $T^{\wedge}\text{Curie}$ " in their manuscript, but " $T^{\wedge}\text{Currie}$ " in Fig. 1d and Fig. 7a,b.

Reviewer #3

(Remarks to the Author)

Rectification effect which can be realized in novel quantum materials without conventional pn junction is now attracting much attention because of its unique functionality as well as its exotic quantum mechanical origins. Especially, when it is realized in superconducting systems, giant nonreciprocal signals including superconducting diode effect can be expected. In this manuscript, authors report the spontaneous voltage generation from rectification of electronic noise in YBCO/NCSMO heterostructure. Large spontaneous voltage (SpV) signals on the order of tens of millivolts can be observed in a rather wide range of temperature and magnetic field. They attributed it to the ratchet-type electronic potential originating from a complex domain state of the NCSMO layers and its proximity effect on YBCO layers. These findings provide a new strategy to realize the rectification effect in quantum materials and also indicates that high T_c superconductor/strong correlated materials heterostructures is an interesting platform for efficient energy harvesting of electronic background noise. I have several questions and comments.

1. Since phase transitions of NCSMO is complex, I think it may be kind for readers of other research field to show a schematic phase diagram of this materials in Supplementary Information.
2. I think controlling the sign of SpV is essential for the application. Based on the authors opinion on the mechanism of SpV, do authors have any ideas or suggestions about controlling the sing of SpV in this system (by making artificial structure or strain gradient etc.)?
3. I am curious to know what is the origin of the electric noise. It is written that "source of energy is provided by the environmental electronic fluctuations/noise that the sample is exposed to inside the magneto-cryostat of a physical property measurement system (PPMS)." Did authors check that SpV will be suppressed when the magnet power supply of PPMS is turned off?
4. I am interested in the difference between stripe contact and point contact. It seems that in the device with point contact, SpV does not show sign change (Fig. 3). What is the reason? Did authors check the magnetic field dependence of this device? (The sign change does not occur even under the magnetic field?)
5. Do authors have any idea about the domain size of polar CCO and FM phase, which are schematically drawn in Fig. 7? How large they are compared with the size of superconducting vortex?
6. I think the observed rectification effect is time-reversal-even phenomena (which does not need external magnetic field and shows symmetric as a function of the magnetic field). I would like to point out the recent related works on similar time-reversal-even nonreciprocal superconducting transport (Nature 604, 653–656 (2022), Nature Communications, 13, 1659 (2022), Nat. Phys. 18, 1221 (2022), Science Advances, 10, eado1502 (2024)) for reference.

Version 1:

Reviewer comments:

Reviewer #1

(Remarks to the Author)

The authors have addressed all my questions and coments. They have also performed additional experiments with a noise inducer to try to reproduce the effect of electrical noise and get a SpV effect (which they found). Hence, I recommend this work to be published in Nature Communications.

Reviewer #2

(Remarks to the Author)

This work is interesting and innovative enough to be published in Nature Communications. I have no further comment.

Reviewer #3

(Remarks to the Author)

In the response letter, authors answered all my questions satisfactorily and revised the manuscript. I feel it is now suitably improved.

Response to the comments by reviewers

We would like to thank all reviewers for their constructive criticism and their interesting questions which have helped us to further improve our manuscript. We would also like to thank them for their overall very positive evaluation of the importance and relevance of our work.

In the following we respond to their criticism and describe the changes that we have accordingly made to the manuscript, in the order of the referee reports.

REVIEWER COMMENTS

Reviewer #1 (Remarks to the Author):

In this work, the authors report the appearance of a spontaneous voltage without applied current in YBCO/NCSMO multilayers. They found that this voltage can be heavily affected by magnetic field and temperature. They propose a possible explanation for this effect based on electronic noise rectification and the nucleation of non-polar FM domains below the Curie temperature of the material, competing with polar COO domains. I have the following remarks concerning this work:

1) In figure 1d it would be necessary to add the magnetization curve for 0.5T, as it is one of the fields used in figure 2a.

We have added the data corresponding to $B=0.5T$ in Fig. 1d.

2) The magnetic field is applied in plane in Figure 1, and I suppose the direction is the same for figure 2, but it could be better to mention it in the text.

The orientation of magnetic field is now mentioned in the figure as well as in the caption.

3) The temperature at which the SpV starts is around 120K, as shown in figure 2b, and appears to be negative and of the order of 0.5mV. Nevertheless, the larger effect, which can reach -80mV starts to appear below the T_c of YBCO, around 80K (which can also be seen in figure 2b). To me the SpV appears to have its origin in the NCSMO, but it is heavily increased when there are superconducting layers. Have the authors tried to measure different samples changing either the NCSMO for other magnetic material or the YBCO for other non superconducting material? For example, multilayers of LSMO/YBCO or multilayers of NCSMO/LaNiO₃. This could clearly point towards the main contribution to

the effect, either the magnetic (and polar) properties of the NCSMO or the superconducting state.

We would like to emphasize that both effects seem to be necessary to obtain a measurable SpV and SpC signal.

We have previously studied various YBCO/La_{2-x}Ca_{1/3}MnO₃ and YBCO/LaMnO₃ multilayers, for which YBCO is superconducting and La_{2-x}Ca_{1/3}MnO₃ is a FM metal and LaMnO_{3-d} can be made insulating or conducting, see e.g. Refs. Malik *et al.*, Phys. Rev. B **85**, 054514 (2012), Marozau *et al.*, Phys. Rev. B **89**, 174422 (2014). For these heterostructures we have not observed any sign of a SpV effect. Clearly, this SpV effect occurs only in heterostructures for which the manganite layer host a complex domain state with the polar COO and the non-polar FM phases, i.e. for the NCSMO studied here or for the Pr_{0.5}La_{0.2}Ca_{0.3}MnO₃ (PLCMO) that we studied previously (Khmaladze *et al.* Phys. Rev. Materials **3**, 084801 (2019)). This highlights that this complex domain state is essential and at the heart of the ratchet-type potential which causes this SpV effect.

On the other hand, we have verified with corresponding experiments on a bare NCSMO layer (as shown in the Supplementary Information) that the manganite on its own does not enable a measurable SpC signal. This appears natural since NCSMO itself is insulating and thus cannot provide free charge carriers capable of sustaining a steady flow to drive current in an external circuit.

This is where the metallic/superconducting YBCO layers appear to be playing an important role. If they are sufficiently thin, they will partially (but not completely) screen the electric field which emanates from the NCSMO layers and thus create a suitable shallow potential landscape for which a measurable charge flow can be activated via the absorption of electronic noise. The thickness of this YBCO layer thus is a critical parameter, as is evident from Fig. 6 of our manuscript. If the YBCO layer is too thick, it can fully screen the electric fields and thus create a short between the contacts. If it is too thin, there will be hardly any screening and thus no measurable current between the contacts. We have verified this for the most extreme case that the YBCO layers are absent, i.e. we have performed SpV and SpC measurements on a single NCSMO layer. The results are shown in the Supplementary Information in Supplementary Fig. 6 and confirm that there is no measurable effect.

We have no data at hand on the SpV effect in corresponding NCSMO/LaNiO₃ heterostructures. However, we would like to mention that YBCO is quite special due to its layered structure and its closeness to a Mott-Hubbard-type insulating state. Both features are probably responsible for its rather large screening length. For LaNiO₃, which is isotropic and not as close to a Mott-Hubbard-insulator state, we thus expect that the screening length is much shorter. Accordingly, very thin LaNiO₃ layers would be required for which the transport properties can be strongly affected by the interfacial disorder and the lateral roughness of these multilayers. This is especially true since LaNiO₃ is close to

a metal-to-insulator transition that is controlled by minor structural changes. In short, while the experiment proposed by the reviewer sounds very interesting, there is a high chance that it would not yield results with straight forward interpretation. There are also other metallic perovskites, like SrRuO₃ or SrVO₃, that could be tested in the framework of such heterostructure with NSCMO, but they would also require a large amount of additional work (for their growth and characterization) that is beyond the scope of our present manuscript.

4) The authors comment about figure 2c are: “Fig. 2(c) reveals that this suppression of the SpV does not happen gradually but rather occurs in two major steps around fields of about 2 and 4 Tesla. At the first step slightly below 2 Tesla the SpV signal is reduced by about an order of magnitude.” From figure 2c what I observe is that there is a change in the sign of the SpV below 2T, not an order of magnitude decrease. I also observe a gradual decrease of the SpV from 3 to 4T (goes from more than 5mV to -1mV). Maybe it is a problem of the scale (only between 5 and -5 mV) or maybe it is a problem of the colours (the same ones for -1 to -3 mV), but I do not agree with the author’s comment about the figure.

We thank the reviewer for pointing out this issue, which is indeed due to a scale problem of the contour plots.

The reviewer will find that we have improved the Figure 2c accordingly. We have modified the scale to indicate >5mV and <5mV to avoid confusion. Adding too many subsection in the color plot is leading to incomprehensibility of the data and hence is avoided.

5) In figure 3, the maximum value of the SpV is 20mV at 50K. Nevertheless, looking at figure 2, there should be maximum (absolute) values of up to 80mV. Moreover, I would expect that the maximum change in SpV in function of field would be around 20-30K, where the voltage changes from -80mV to 0mV according to figure 2. I understand that in one case the V vs T curves are measured with continuously applied field, while the V vs H not, but why is the behavior so different between figure 2 and 3?

Our response is twofold:

Firstly, we do not show in Fig. 3 the full virgin curve at 10K, which starts at a much larger SpV signal. Also note that the loops at higher temperatures have been subsequently performed by stepwise increasing the temperature. Accordingly, these SpV signals cannot be directly compared to the SpV-T curves in Fig. 2a which were taken during cooling in different fixed magnetic fields.

Secondly, we find (and have also pointed out in the paper) that the SpV and SpC signals are on a quantitative level not reproducible during successive runs that start from elevated temperatures (above the COO transition around 200K). There is also a strong history dependence with respect to sweeping the magnetic field. What is reproducible, however, are the qualitative features that we are describing in our manuscript.

6) In figure 4, could it be possible to present also the curves for -10uA? I do not understand how in the case of $T=50\text{K}$, for example, the voltage of the multilayer is lower when applying 10uA than with 0uA. In principle, and looking at figure 1c, all the current should flow through the YBCO layers, giving a zero resistance state. How can the current flowing through the YBCO affect the SpV effect in the NCSMO layers? It could be interesting to show current-voltage curves to have a more general view of the effect of applied currents. Also, when the authors say they apply 0uA, it is with the current source on, applying 0uA or with the current source off?

We would like to point out that the domain state in NCSMO appears to be rather versatile, since it is determined by a delicate balance between the various competing and nearly degenerate phase, like the insulating polar/AF, the insulating non-polar/AF, and the conducting non-polar/FM. In particular, the latter is favored by a magnetic field and likely also by an applied current.

Figure 1: I-V characteristics of a NYN Superlattice at different temperatures. Please note that at $T < 150\text{K}$, $I=0$ does not correspond to $V=0$.

Note that if we are cooling down from elevated temperature (above this domain state) under an applied current, the latter most likely influences the domain formation in the

NCSMO layer such that the blockage due to the insulating and polar layers is less pronounced.

In Figure 1 we show IV curves for the SL which suggest that one of the YBCO layers indeed becomes superconducting SC and carries the applied current, whereas the other YBCO and NCSMO layers remain charge locked and thus maintain a voltage offset that persists at zero applied current.

7) Figure 6 shows the maximum amplitude of the SpV in function of YBCO thickness. Following the trend, have the authors tried to use a YBCO thin enough (1 or 2 unit cells) so that the YBCO is insulating? What would the authors expect to happen then?

We have performed corresponding SpV and SpC effect measurements on a plain NCSMO layer, as shown in Supplementary Fig. 6 (b) and (c) in the Supplementary Information. Here we find no detectable SpV or SpC effect. This is likely because the carriers in NCSMO are not mobile enough to sustain a measurable lateral current (on a macroscopic scale). Similar result can be expected for multilayers with 1 or 2 unit cell thick YBCO layers that would be insulating. We have not grown such samples (yet) and performed such measurements. The reason is that we expect additional complications due to the additional problem that the roughness of the layers becomes comparable here to the thickness of the YBCO layer.

8) With respect to the SpV originating from environmental electronic noise, could the authors purposely introduce noise, for example with an RF antenna, to controllably increase the SpV effect?

Figure 2: Measurement setup indicating the noise inducing coil -set and the sample inside the PPMS cryostat

Following the interesting suggestion of the reviewer, we have performed such an experiment. For this we have constructed a separate insert for our PPMS system which allows to place a noise inducing coil close to the sample, as shown in Supplementary Fig. 4. This coil has been connected with an ac function generator that provides a broadband (50MHz) white noise to the sample, while it is cooled down to different measurement temperatures. Two probes of a digital multimeter were connected to two line contacts on the sample (in the same geometry as the measurement of SpV). Supplementary Fig. 4 (b) shows that there is large difference in the SpV between the 'ON' and 'OFF' states of the function generator. Further, this difference is a function of the sample temperature as well as the

amplitude of the applied noise voltage. Further details are discussed in the revised Supplementary Information in section 4.

9) With respect to the discussion, how do the authors explain that, for example, for the SpV measured at 2T, the SpV starts to appear at 120K while magnetization measurements show that the system is magnetic at temperatures as high as 225K? According to the discussion, the presence of magnetic domains (and hence, the absence of polar domains) is responsible for the SpV

According to our interpretation, there are two conditions that must be met for the SpV and SpC effect to occur.

- 1.) There should be some non-polar FM domains that replace parts of the polar AF domains.
- 2.) There should not be too many of these non-polar FM domains. Otherwise, they can develop percolating pathways along which the electric fields (that emanate from the domain boundaries of the NCSMO layers) are weak and can be fully screened by the YBCO layers. This results in metallic/superconducting lateral pathways which can short SpV and SpC. This is expected to happen at high magnetic fields where the fraction of the non-polar FM phase is strongly enhanced.

In short, the proposed mechanism requires some non-polar FM domains, but it is suppressed when too many of them are present, e.g. at high magnetic field!

Small changes:

- In line 70 CE-type AF is not defined (Charge-Exchange-type)

According changes have been made to the revised manuscript in line no. 67.

- The references 18 to 21, which the authors cite in the introduction to talk about $(\text{Pr,Nd})_{1-x}(\text{Ca,Sr})_x\text{MnO}_3$, only refer to $\text{Pr}_{1-x}(\text{Ca,Sr})_x\text{MnO}_3$ and other manganites, but not to $\text{Nd}_{1-x}(\text{Ca,Sr})_x\text{MnO}_3$, which is the material used in this work. They should cite bibliography about this specific material, or at least a manganite with Nd.

The complex phase diagram of the manganites is governed by the average size of the A site ions. The general trend is that small rare ions, like Pr and Nd, both tend to stabilize the COO-phase, the smaller Pr $3+$ a bit more than the Nd $3+$. Actually, the larger difference is due to the Sr $2+$ and Ca $2+$ ions. In the literature one finds indeed much more information about PrCaMnO_3 than about the NdCaMnO_3 . However, our Ref. 18 (Tokura, Rep. Prog. Phys. **69**, 797(2006)) shows in Fig. 11 that the two give very similar COO state for the case of 50% Ca doping. Moreover, in our previous work we have observed similar SpV effects for $\text{Pr}_{0.5}\text{La}_{0.2}\text{Ca}_{0.3}\text{MnO}_3$ layers (see Ref. Mallett Phys. Rev B **94**, 180503(R) (2016)) and

shown that they are entirely absent with $\text{Nd}_{0.3}\text{Sr}_{0.7}\text{MnO}_3$ which is in a FM metallic state (Khmaladze et al Phys. Rev. Materials **3**, 084801 (2019)).

- In figure 3 the color code should also be written in the figure text to make it clearer. Also in figure 3f, the minus sign of the SpV cannot be seen properly.

The color code for showing the successive field sweeps in Fig. 3 is now explicitly mentioned in the figure caption.

The axis title has been shifted to the left for better legibility.

- Lines 176 to 178 indicate that the scenario of a competition between different phases in NCSMO has been described already, but it is explained later in the paper. We have removed “above described”.

In summary, the work presented here is original, scientifically valid (although the origin of the effect may be subject to doubt). It is an interesting work, but I think its relevance can be limited to scientists working outside of the complex oxides field, as the properties needed for this effect can only be found in this type of materials. Therefore, I would like my remarks to be addressed before I can recommend it for publication in Nature Communications.

We thank the reviewer for concluding that our work is “original”, “scientifically valid” and “interesting”.

Concerning the relevance of our work to scientist working outside of the complex oxide field we would like to remark the following. Our work highlights that interesting new and potentially useful properties and functionalities can be obtained from materials with competing orders. This concept finds many applications in a wide range of disciplines, including life science where it enables, e.g. the folding of molecules. Focusing primarily on condensed matter material science, our work may motivate the search for heterostructures made from various other types of materials with competing polar electric and magnetic orders. Moreover, it will hopefully inspire further research into the versatile properties of these complex oxides. Also, the study has a potential application in the field of sensors for detecting ambient signals, for example in space research.

In addition to these facts, an important contribution of our work is the demonstration of time-reversal-symmetric rectification effect. There is currently a great interest in the study of non-reciprocal phenomena. In most cases, this arises from magnetochiral anisotropy which results in anti-symmetric behavior with respect to magnetic field polarity. Consequently, the effect vanishes at zero field. Only very few examples (Refs. 42-46) are known where non-reciprocity is time-reversal-symmetric and persists at zero field. Our

work adds to this, and is unprecedented from the perspective that both non-reciprocity and spontaneous voltage are observed. The temperature range of more than 100 K is unique in current literature on this this topic. Thus, this work is of relevance as a model system to the entire field of non-reciprocal transport, beyond the complex oxide community.

Reviewer #2 (Remarks to the Author):

In this article, Soulier et al. stacked an artificial cuprate/manganite-stacked superlattice structure, and investigated the electronic transport properties of such a structure. Apart from the expected superconducting transition of YBCO and magnetic transitions of NCSMO layers, it is quite interesting that a spontaneous voltage signal below ~130 K was observed, with an amplitude being as large as tens of mV. Such spontaneous voltage signal can be modulated by external magnetic field due to the modulation of domain structure of NCSMO layers, and shows potentials for memory functionality. Generally, I think this work is interesting and innovative enough to be published in Nature Communications. But still, there are some points that I think the author should consider prior to the acceptance of this work.

We would like to thank the reviewer for his positive evaluation of our work.

1. I found that the Introduction section is kind of too long to catch the main parts that the authors intended to introduce. Therefore, I suggest the authors shorten their Introduction section and highlight the key points.

The reviewer will find that we have substantially shortened the introduction part of the revised manuscript.

2. The heterostructure is similar to the Ba-Nb-S system where the superconducting layer is sandwiched by non-superconducting materials (Devarakonda *et al.*, Science **370**, 231 (2020)) where 2D superconductivity and FFLO state was claimed. Similarly, can the authors observe any feature of superconducting YBCO in such a heterojunction that is different from the bulk YBCO samples?

We thank the reviewer for pointing out this interesting work. On a purely phenomenological level there are indeed some parallels, like the finding that a coherent SC response can be restored by large magnetic fields. However, the underlying mechanism is most likely very different. In our case, it is not an intrinsic property of the YBCO layer itself, but rather a consequence of a field-induced enhancement of the fraction of the FM phase in the manganite layer that reduces the electric stray fields into

the YBCO layer which thus can be fully screened to restore a coherent SC state. See also our response to reviewer 1 under his point 3.

The reviewer will find that we have added a corresponding discussion in the conclusion part of our manuscript.

3.The orientation of magnetic field (in-plane or out-of-plane) should be clarified in Fig. 2. A schematic illustration is suggested.

The requested information about the field orientation has been added to the caption of Fig. 2. A schematic illustration is shown in Fig. 1a.

4.Are such observations of SpV expected to be reproduced based on FM materials with out of-plane magnetic easy axis? For instance, the insulating FM material, Cr₂Ge₂Te₆, can have multi-domain structure with net magnetization ~ 0 (similar to the illustration in Fig. 7 in this manuscript).

The Cr₂Ge₂Te₆ system does not seem to be directly comparable, since it is lacking the polar domains which are essential to our proposed mechanism.

5.I am quite curious the role Josephson effect plays in such an SpV effect, since the superconducting order parameter between adjacent YBCO nanoflakes should form a weak-coupling Josephson junction. Since 0-junctions and/or pi-junctions can form in SC/FM/SC heterostructures, I am wondering if these junctions exist in such a heterostructure. Will they affect the electronic properties of the heterojunction?

This is a very interesting question which deserves further attention but is beyond the scope of our present manuscript. The regions where the YBCO layer is not thick enough to fully screen the electric field (emanating from the NCSMO layer) are most likely in a state close to the undoped Mott-Hubbard insulator, and thus may act as Josephson-type junctions and complex networks thereof. They could indeed be playing an important role in the observed large jumps in the magnitude and sign of the SpV signal.

In the revised manuscript, we have added a short discussion in the conclusion and outlook section where it now says: “The latter might also act like networks of Josephson-type junctions that could explain the memory effects in the magnetic field loops shown in Fig. 3.”

6.The authors used “ T^{\wedge} Curie” in their manuscript, but “ T^{\wedge} Currie” in Fig.1d and Fig. 7a,b.

It has been corrected in the manuscript.

Reviewer #3 (Remarks to the Author):

Rectification effect which can be realized in novel quantum materials without conventional pn junction is now attracting much attention because of its unique functionality as well as its exotic quantum mechanical origins. Especially, when it is realized in superconducting systems, giant nonreciprocal signals including superconducting diode effect can be expected. In this manuscript, authors report the spontaneous voltage generation from rectification of electronic noise in YBCO/NCSMO heterostructure. Large spontaneous voltage (SpV) signals on the order of tens of millivolts can be observed in a rather wide range of temperature and magnetic field. They attributed it to the ratchet-type electronic potential originating from a complex domain state of the NCSMO layers and its proximity effect on YBCO layers. These findings provide a new strategy to realize the rectification effect in quantum materials and also indicates that high T_c superconductor/strong correlated materials heterostructures is an interesting platform for efficient energy harvesting of electronic background noise.

I have several questions and comments.

1. Since phase transitions of NCSMO is complex, I think it may be kind for readers of other research field to show a schematic phase diagram of this materials in SOM.

As requested by the reviewer, a schematic phase diagram of the manganites is now shown in Supplementary Fig. 1 of the Supplementary Information.

2. I think controlling the sign of SpV is essential for the application. Based on the authors opinion on the mechanism of SpV, do authors have any ideas or suggestions about controlling the sign of SpV in this system (by making artificial structure or strain gradient etc.)?

This important point is now mentioned in the outlook section of our revised paper.

It is indeed an interesting question. As shown in Fig. 3, one can also change the sign of the SpV signal with an applied magnetic field. Further studies how these reversible jumps in the SpV-H loops depend on strain, defects etc. will thus be highly desirable.

3. I am curious to know what is the origin of the electric noise. It is written that “source of energy is provided by the environmental electronic fluctuations/noise that the sample is exposed to inside the magneto-cryostat of a physical property measurement system

(PPMS).” Did authors check that SpV will be suppressed when the magnet power supply of PPMS is turned off?

We have been advised by Quantum Design that it is not safe to switch off the magnet power supply during measurement operations.

We have therefore instead built a separate insert for the PPMS sample chamber which allows us to expose the sample to a controlled electromagnetic noise. The outline of the setup and the obtained results are now shown in the Supplementary Information in Supplementary Fig. 4? and Section 4. The reviewer will find that the SpV signal is indeed sensitive to the power and the frequency of the electromagnetic radiation.

4. I am interested in the difference between stripe contact and point contact. It seems that in the device with point contact, SpV does not show sign change (Fig. 3). What is the reason? Did authors check the magnetic field dependence of this device? (The sign change does not occur even under the magnetic field?)

The wire-bonded point contacts cover only around 60 micron x 60 micron area and they reach through the thin NCSMO layer at the surface. The invasive nature of the point contacts ensures a coupling to all the YBCO layers of the superlattice but it may also affect the physical properties in their neighbourhood. This makes the point contacts quite complex to understand. We have therefore carried out most of the measurements with the line contacts. In particular, the magnetic field measurements have been mostly performed on samples with line contacts and not on samples with wire-bonded contacts.

Although there is no sign change in SpV shown in Fig. 5 (with point contacts), we have seen a sign changes between successive experimental runs. We show here (Response Fig. 3) data from two samples with wire-bonded point contacts. Each figure shows SpV as a

Figure 3: Sign change of the SpV under repeated thermal cycles for (a) a NY-SL and (b) a NYN trilayer with wire-bonded point contacts.

function of temperature at zero magnetic field for two different thermal cycles (during which the experimental setup was not modified). The sign of the SpV can be seen to change between the two cycles.

5. Do authors have any idea about the domain size of polar CCO and FM phase, which are schematically drawn in Fig. 7? How large they are compared with the size of superconducting vortex?

The only direct information is from the TEM study of Jooss, C. et al. PNAS **104**, 13597-13602 (2007). They have shown that the CCO domains have a size on the order of about 20 nanometer (this is also stated in our manuscript). The domain size is thus about an order of magnitude larger than the SC coherence length and an order of magnitude smaller than the SC penetration of (bulk) YBCO.

Unfortunately, we do not have further information at hand on the interesting question of how the SC vortices in YBCO interact with this complex domain state in NCSMO. Hopefully some direct, microscopic studies of the vortex distribution will be motivated by our present work.

6. I think the observed rectification effect is time-reversal-even phenomena (which does not need external magnetic field and shows symmetric as a function of the magnetic field). I would like to point out the recent related works on similar time-reversal-even nonreciprocal superconducting transport (Nature 604, 653–656 (2022), Nature Communications, 13, 1659 (2022), Nat. Phys. 18, 1221 (2022), Science Advances, 10, eado1502 (2024)) for reference.

The observed effect is indeed a time-reversal-even phenomenon. This is verified both by the even symmetry with respect to the magnetic field polarity (Fig 3) and the spontaneous voltage without any applied field (Fig. 2(b)).

Our work is distinguished from previous reports [1-4] by the observation of a large SpV over a wide range of temperatures. As far as we are aware, this has not been reported earlier. The only instance where SpV with even-in-field symmetry was reported [5] involves disordered Nb channels. However, even in that case, SpV at zero field appears only in a very small temperature window around T_c (less than 3 K). Our observation regarding the SpV spanning a range of temperatures more than 100 K is quite unprecedented.

The reviewer will find that we have included the above discussion and references in the conclusion and outlook section of the revised manuscript.

[1] Nature 604, 653–656 (2022)

[2] Nature Communications, 13, 1659 (2022)

[3] Nat. Phys. 18, 1221 (2022)

[4] Science Advances, 10, eado1502 (2024)

[5] Physical Review B 109, L060503 (2024)